# A long-term proxy for sea ice thickness in the Canadian Arctic: 1996-2020

Isolde A. Glissenaar[1], Jack C. Landy[2], David G. Babb[3], Geoffrey J. Dawson[1], and Stephen E.L. Howell[4]

[1]Bristol Glaciology Centre, School of Geographical Sciences, University of Bristol, Bristol, UK
[2]Centre for Integrated Remote Sensing and Forecasting for Arctic Operations, Department of Physics and Technology, UiT The Arctic University of Norway, Tromsø, Norway
[3]Centre for Earth Observation Science, University of Manitoba, Winnipeg, MB, Canada
[4]Climate Research Division, Environment and Climate Change Canada, Toronto, ON, Canada

**Correspondence:** Isolde A. Glissenaar (isolde.glissenaar@bristol.ac.uk)

**Abstract.** This study presents a long-term winter sea ice thickness proxy-product for the Canadian Arctic based on a Random Forest Regression model - applied on ice charts and scatterometer data, trained on CryoSat-2 observations, and applying an ice type-sea ice thickness correction using PIOMAS - that provides 25 years of sea ice thickness in the Beaufort Sea, Baffin Bay, and, for the first time, the Canadian Arctic Archipelago. An evaluation of the product with in-situ sea ice thickness measurements shows that the presented sea ice thickness proxy product correctly estimates the magnitudes of the ice thickness and accurately captures spatial and temporal variability. The product estimates sea ice thickness within 30 to 50 cm uncertainty from the model. The sea ice thickness proxy-product shows that sea ice is thinning over most of the Canadian Arctic, with a mean trend of -0.82 cm/year in April over the whole study area (corresponding to 21 cm thinning over the 25-year record), but that trends vary locally. The Beaufort Sea and Baffin Bay show significant negative trends during all months, though with peaks in November (-2.8 cm/yr) and April (-1.5 cm/yr), respectively. The Parry Channel, which is part of the Northwest Passage and relevant for shipping, shows significant thinning in autumn. The sea ice thickness proxy product gives, for the first time, the opportunity to study long-term trends and variability in sea ice thickness in the Canadian Arctic, including the narrow channels in the Canadian Arctic Archipelago.

## 1   Introduction

Sea ice thickness (SIT) is a key variable when characterising an ice cover and its impact on the local environment, and provides important insight into how an ice cover is changing in response to climate change. Unfortunately, observations of ice thickness at appropriate spatial and temporal scales are sparse. Seasonal estimates of ice thickness from satellite altimeters only go back to 2003, while year-round observations only extend back to 2010 (Landy et al., 2022), and represent a rather short record for examination of long-term trends and variability. In place of observations, reanalyses such as the Pan-Arctic Ice Ocean Modeling and Assimilation System (PIOMAS, Zhang and Rothrock, 2003) are commonly used to provide long-term estimates of ice thickness, however PIOMAS is known to overestimate thinner ice and underestimate thicker ice (Schweiger et al., 2011). Furthermore, both satellite altimeters and PIOMAS have difficulty resolving ice thickness in coastal areas and either mask out

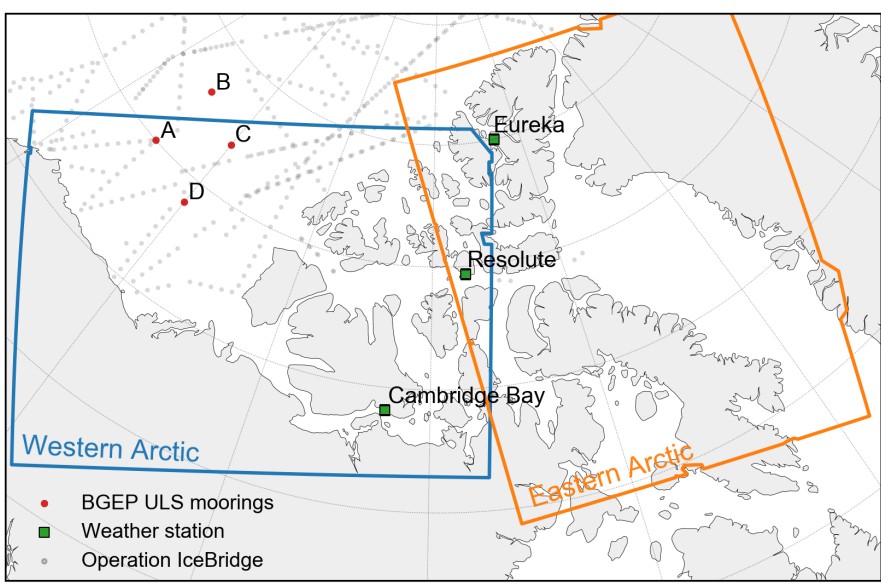

**Figure 1.** Overview of the study area with locations of the Canadian Ice Service regional ice charts for the Western Arctic and Eastern Arctic (Canadian Ice Service (CIS), 2007a), the BGEP ULS moorings (Krishfield and Proshutinsky, 2006), the weather stations for measuring land fast ice thickness (Brown and Cote, 1992), and Operation IceBridge flight paths (Kurtz et al., 2015).

or have a high degree of uncertainty over the Canadian Arctic Archipelago (CAA). Satellite altimeters are limited in their application within the CAA because of a combination of a lack of leads within the seasonally landfast ice cover, strong tidal cycles and a lack of snow depth products, which collectively result in large uncertainties in sea ice freeboard (Ricker et al., 2014). PIOMAS has a high degree of uncertainty within the CAA due to mix of seasonal and multi-year sea ice (Howell et al., 2016). As a result, observations of ice thickness within the CAA are confined to few opportunistic observations (i.e., Melling, 2002; Haas and Howell, 2015; Melling et al., 2015). Despite the difficulty in observing ice thickness within the CAA, it is estimated to contain about 10% of the northern hemisphere sea ice volume (Lietaer et al., 2008) and is home to some of the oldest and thickest MYI in the Arctic (Bourke and Garrett, 1987; Barber et al., 2018; Haas et al., 2010; Melling, 2022). The CAA is an important pathway bringing cold, fresh Arctic water to the Labrador Sea (Melling et al., 2008), which is an important site for deep convection and plays a key role in the large-scale meridional overturning circulation (e.g., Marshall and Schott, 1999). Moreover, the CAA is bisected by the Northwest Passage and is home to many northern communities that rely on maritime traffic for resupply (Dawson et al., 2020). As the ice cover declines, ship traffic across the Canadian Arctic has dramatically increased since the 1990s (Pizzolato et al., 2014) and sea ice poses the greatest risk to ships operating within the CAA. Understanding the changes in ice thickness within the CAA and monitoring it in the future is therefore of vital importance.

Here, we combine information from the Canadian Ice Service (CIS) ice charts and aggregated scatterometer backscatter data to create a proxy SIT product over the Canadian Arctic, including the Beaufort Sea, Baffin Bay and the CAA (Figure

1), for November-April from 1996-2020. We apply machine learning methods on these long-term remote sensing datasets and CryoSat-2 SIT observations to determine the relationship between sea ice stage of development, form of ice, backscatter and SIT to create the SIT proxy-product. Additionally, this machine learning model can be used moving forward to provide estimates of ice thickness, whenever ice charts and scatterometer imagery are available. Within the paper Section 2 introduces the datasets and Section 3 the applied methods. Section 4 evaluates the proxy-product versus satellite and in-situ observations of SIT. Section 5 presents the results of the proxy-product and discusses the emergent trends and variability of SIT in the Canadian Arctic.

## 2 Data

### 2.1 Canadian Ice Service Ice charts

The CIS has produced ice charts for the Canadian Arctic (Figure 1) since the 1960s that include information on sea ice concentration, stage of development (relating to age, ranging from new ice to multi-year ice), and form of ice (relating to floe size or kind of ice (e.g., iceberg, fast ice)) using the World Meteorological Organization egg code. The ice charts use polygons to represent different ice regimes, consisting of up to three different stages of development and forms of ice. The uncertainty in the ice charts has been reviewed and the data was validated for use in climate studies (see Canadian Ice Service (CIS), 2007a; Tivy et al., 2011). Numerous studies have used the ice charts to study trends and variability in sea ice cover (Tivy et al., 2011; Mudryk et al., 2018; Derksen et al., 2018), to quantify loss of multi-year ice in the Canadian Arctic (Galley et al., 2016; Babb et al., 2022; Howell et al., 2022), and to research causes for sea ice extremes (Howell et al., 2010; Babb et al., 2019, 2020). The ice charts are produced by ice analysts who compile available aerial, shipping, and remote sensing data, though since 1996 RADARSAT has been the primary data source (Canadian Ice Service (CIS), 2007b; Tivy et al., 2011).

Previous studies have shown that there is a direct relationship between ice age and SIT and have used this relationship to propose simple linear models that derive SIT for March (Maslanik et al., 2017; Tschudi et al., 2016; Liu et al., 2020). Additionally floe size has been shown to be related to ice age and ice thickness as thicker MYI floes tend to be larger than FYI floes (Tilling et al., 2019; Hwang et al., 2017; Aldenhoff et al., 2019). The variables stage of development (from here on called ice type) and form of ice in the CIS ice charts are related to observed SIT (Figure 2) which gives the possibility to use the ice charts to estimate ice thickness. The stages of development used in the ice charts are provided with an estimated range of SIT. However, the actual relations between the stages of development, form of ice, and SIT are currently unknown.

The weekly regional ice charts have all been digitised and are freely available at https://iceweb1.cis.ec.gc.ca/. This study uses the ice charts for the Western Arctic and Eastern Arctic for November-April 1996-2020, since the start of the ice analysts using RADARSAT as primary data source (Figure 1). The temporal availability of the ice charts varies from monthly to weekly (see Supplementary Table S1) and changes in spatial coverage slightly in 1997 and 1998.

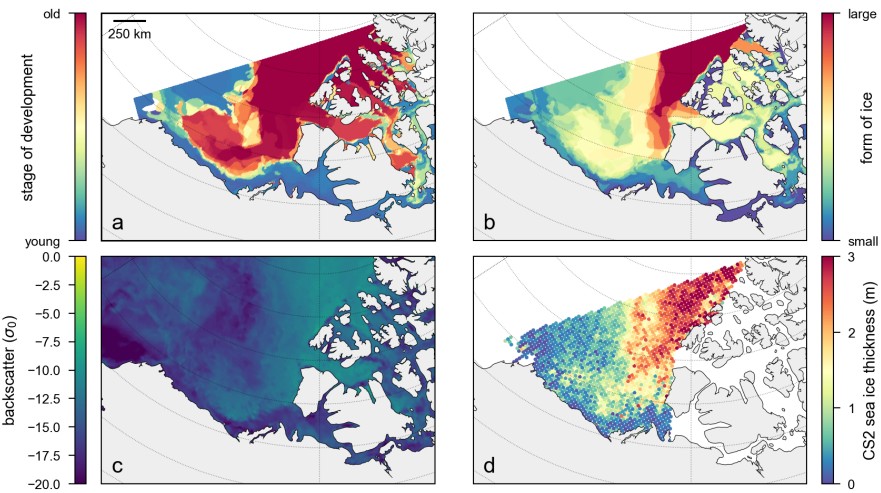

**Figure 2.** One snapshot of the remote sensing input data compared to observed sea ice thickness in November 2018 with (a) stage of development from the CIS ice charts, (b) form of ice from the CIS ice charts, (c) scatterometer backscatter from ASCAT, and (d) CryoSat-2 observed sea ice thickness.

## 2.2 Scatterometer data

Within the CIS charts polygons the mix of stages of development and forms of ice is assumed to be uniform, whereas we expect sea ice to be more spatially variable on a smaller scale. Because of this, the dataset was supplemented with scatterometer image reconstruction (SIR) sigma-naught calibrated backscatter data (Early and Long, 2001) from multiple scatterometer satellites. Scatterometer backscatter records, going back to 1992, have previously been used to create ice age products (e.g., Zhang et al., 2019; Lindell and Long, 2016) and are suggested by Belmone Rivas et al. (2018) as a reliable proxy in the historical reconstruction of SIT due to its spatial correlation with observed ice thickness. As there is no continuous record of one instrument over the entire 1996-2020 record, we use data from multiple satellite scatterometers that operate in the C-band (4-8 GHz) and Ku-band (12-18 GHz) (Figure 3). Each of the scatterometers used in this study are detailed in Appendix A.

After comparing the different scatterometer data products, we determined that there was no bias between the different sensors operating in the same band, and combined the datasets into a long-term record for each wavelength. As the wavelengths interact differently with snow and ice (Ontstott, 1992), we expect C-band and Ku-band instruments to give different results and do not combine scatterometer data from the different bands into one record. C-band scatterometer data from ERS-1, ERS-2, and ASCAT was combined into one record from 1996-2020 with a gap from 2001 to 2007. Ku-band scatterometer data from QuickScat, OSCAT-1, and OSCAT-2 was combined into one record from 1999-2020 with a gap from 2014 to 2017. More detail on the satellites, an analysis of the data, and justification for using the long-term time series without bias correction is provided in Appendix A.

**Table 1.** Details on scatterometer data.

|  | Time period | Frequency | Launched by |
|---|---|---|---|
| ERS-1[1] | Jan 1996 - Apr 1996 | C-band | ESA |
| ERS-2[1] | Nov 1996 - Jan 2001 | C-band | ESA |
| ASCAT[1] | Jan 2007 - Dec 2020 | C-band | ESA |
| QuickScat[1] | Nov 1999 - Nov 2009 | Ku-band | NASA |
| OSCAT-1[1] | Nov 2009 - Feb 2014 | Ku-band | ISRO |
| OSCAT-2[2] | Nov 2016 - Dec 2020 | Ku-band | ISRO |

1. Obtained from NASA SCP: https://www.scp.byu.edu/data/

2. Obtained from MOSDAC: https://mosdac.gov.in/satellite-catalog

## 2.3 CryoSat-2

We used seasonal ice thickness measurements derived from ESA's CryoSat-2 radar altimeter for November to April for the period 2010 to 2020 using the methodology described in (Landy et al., 2020). This methodology applies a numerical model for backscattered CryoSat-2 SAR echo waveform, assuming lognormal statistics for the sea ice height and roughness distribution, to retrieve sea ice freeboard and the SnowModel-LG snow depth and density (Liston et al., 2020) to estimate SIT. We use monthly SIT observations on a 50 km grid, however we mask out SIT observations from the Canadian Arctic Archipelago (as defined by MASIE-NH regions (Fetterer et al., 2010)), as we cannot reliably obtain SIT measurements with land contamination of the return echo, a lack of leads reduces the performance of CryoSat-2 and the SnowModel-LG product is not available in this region. We also removed outliers in the data by excluding any SIT measurement with an uncertainty in the top 5th percentile (more than 0.48 m).

## 3 Methods

### 3.1 Creating training dataset

We trained the machine learning model using a dataset from November-April 2010-2020 of predictor features including the partial concentration (between 0 and 1) of each ice type (new ice, nilas, ... multi-year ice) and form of ice (pancake, small ice cake, ... giant floe) from the ice charts and scatterometer data. This resulted in 24 input variables (see Supplementary Table S2 for full list of used predictor features). These data were gridded to the same 50 km resolution as the CryoSat-2 SIT observations, which are used as coincident reference observations to train the model on. Sea ice drift moving ice between grid cells during the month might introduce some uncertainty, especially in the Beaufort Gyre where the drift is high (Petty et al., 2016). However, with the low resolution of 50 km of the dataset, we assume this effect will not be large. As the CryoSat-2

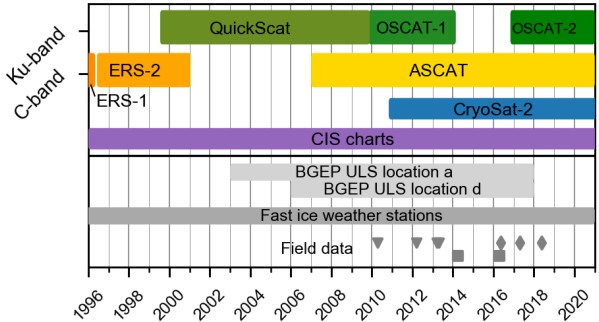

**Figure 3.** Timeline of availability of used data in the machine learning model and in-situ data used for validation. Field data triangles refer to Operation IceBridge campaigns, squares to the ECCC fast ice measurements in Eureka, and diamonds to the fast ice measurements in Cambridge Bay.

SIT observations have been masked out for the Canadian Arctic Archipelago, this region is not included in the training dataset. Nevertheless, the distribution of the features in the training region was after inspection determined to be representative for the full region. Each grid node was taken as an individual data point to be used in the training. Separate training datasets were created for the C-band and Ku-band scatterometer data. As the relationship between ice age and SIT varies over time in the sea ice growth season, the training dataset was separated by month. The number of points in the training datasets varies from 14,642 to 30,601 (full list in Supplementary Table S2).

The categories second-year ice (SYI) and multi-year ice (MYI) were not used in the ice charts for the months of January-April during the training period (2010-2020) but have been used in the ice charts for previous years of the long-term record, therefore we could not use these variables, instead we combined the SYI and MYI features for January-April into the overarching 'Old Ice' feature, which appears consistently in the training period and full record. There are some other rare instances where a feature is used in the long record of predicting features but not in the training dataset. These other features do not have overarching categories, so we decided to remove the sporadic instances where this feature has a partial concentration of more than 50% within a grid cell.

## 3.2 Random Forest Regression model

After comparing with the performance of linear regression, decision tree regression, and gradient boosting regression, a Random Forest Regression model was selected as the most suitable machine learning model for this task. We trained the Random Forest Regression model to find the relation between the predictor features (the ice type, form of ice and backscatter) and observations (CryoSat-2 observed SIT), in order to create a proxy SIT record for 1996-2020. The full processing chain is visualised in Figure 4. Random Forest Regression is a supervised learning algorithm that uses ensemble learning. A Random Forest operates by constructing several decision trees during training and outputting the final predicted value as the mean prediction of all the trees. A Random Forest is a powerful model, capable of finding complex nonlinear relationships in data.

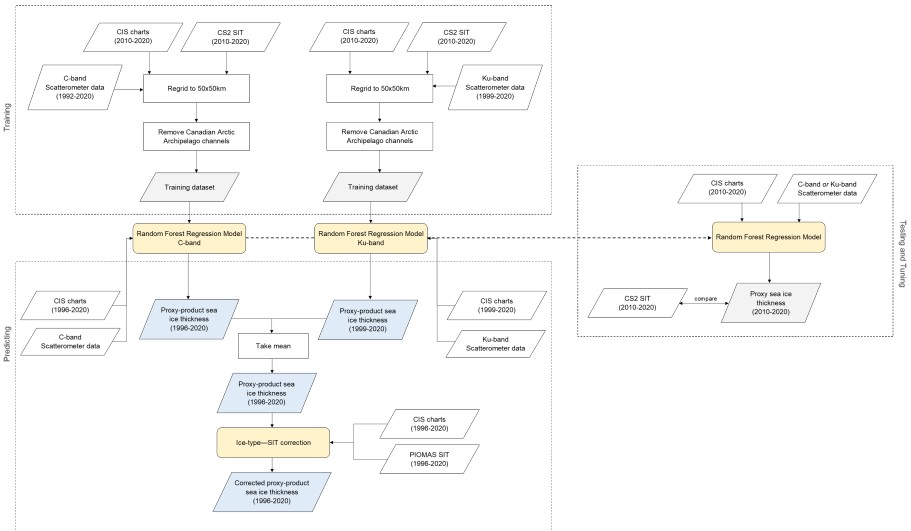

**Figure 4.** Schematic of applied data and methods to create the sea ice thickness proxy product.

The optimal parameters were selected using the Python package scikit-learn's hyperparameter tuning function GridSearchCV. The number of trees in the forest was set at 95, the maximum depth of the separate trees is 15 levels, and the number of features to consider when looking for the best split was set at 5. The other hyperparameters were set at their default. A separate Random Forest Regression model was created for each month in November-April. There are also separate models for the datasets with Ku-band scatterometer data and C-band scatterometer data. The results of these were combined after the SIT prediction was made by taking the mean of the two results where both are available (Figure 3).

The results for December 1996-April 1997 were removed from the analysis of the proxy SIT product as they showed unreasonably high SIT results (>1 m thicker than in other years) caused by the Canadian ice charts showing a full cover of old ice in the Beaufort Sea. This was deemed as highly unlikely, as such a large area appeared very differently, for these months, in all other years of the ice chart record. ERS2 scatterometer backscatter supports the higher MYI concentration in the southern Beaufort Sea in this year, showing higher backscatter. However, ERS2 scatterometer backscatter does not support these extreme conditions in the entire Beaufort Sea in the ice charts as there is no anomaly in ERS2 scatterometer data in the western Beaufort Sea and Central Arctic. We assume it was an overestimation in the interpretation by the ice analyst.

### 3.3 Correction to thinning of ice types

One of the assumptions in the generated SIT proxy-product is that the relation between the inputs (ice type, form of ice, and scatterometer backscatter) and the SIT stay consistent during the period the model is applied (1996-2020). However, we know that over the recent past Arctic multi-year ice has thinned (Kacimi and Kwok, 2022; Krishfield et al., 2014). In order to correct for this change, we retrieved the linear least-squares trend in PIOMAS mean thickness (available http://psc.apl.uw.edu/ research/projects/arctic-sea-ice-volume-anomaly/data/model_grid) of the three coincident overarching categories (multi-year

**Table 2.** PIOMAS trends in sea ice thickness in main ice type categories in cm/yr. No number given when not significant (p>0.1).

|  | Young ice | First-year ice | Multi-year ice |
|---|---|---|---|
| November | - | - | -3.0 |
| December | - | -0.9 | -3.8 |
| January | - | -0.4 | -2.8 |
| February | - | -0.6 | -1.8 |
| March | - | -0.5 | -1.5 |
| April | - | -0.6 | - |

ice, first-year ice and young ice) for the region covered by the ice charts and for the period 1996-2020. The region of PIOMAS data within the Canadian Arctic channels was masked out. Trends are presented in Table 2; there are significant (p<0.05) negative trends for MYI for every month between November and March and for FYI for every month except November and March. For the categories and months where the trend is significant, the trend was applied as a correction to the SIT results in the proxy-product as follows:

$$SIT_{corr} = SIT + t \cdot (trend_{MYI} \cdot C_{MYI} + trend_{FYI} \cdot C_{FYI} + trend_{YI} \cdot C_{YI}) \tag{1}$$

where $SIT$ is the sea ice thickness, $trend$ is the sea ice thickness trend over time within the given ice category, $t$ the time in years prior to 2015 (the middle of the training data period), and $C$ the partial concentration of the given ice category. We present both the raw SIT from the proxy-product, which we refer to as proxy_nocorr, and a product corrected for changes in SIT within specific categories, which we refer to as proxy_corr. The proxy_nocorr SIT product quantifies the sea ice thickness change due to changes in ice type, floes size and scatterometer backscatter, and can be used to study sea ice thickness variability and to apply to newly released ice charts. The proxy_corr product can be used when determining long-term trends in SIT, accounting for both changes due to ice type, floe size and backscatter, and expected changes in the relationship between SIT and ice type. There are significant uncertainties associated with PIOMAS sea ice thickness observations (Schweiger et al., 2011) and we can expect these to be highest within the channels of the Canadian Arctic. Numerical simulation of the sea ice dynamics and ice-ocean exchanges is challenging within such a complex fjord environment and the PIOMAS solution is constrained by satellite data such as passive microwave sea ice concentrations that are also uncertain in this region. For this reason, we mask out the CAA channels when estimating long-term SIT trends by ice type and consider the proxy_corr SIT product more accurate for analysing trends within the Canadian Arctic channels than simply using the PIOMAS data.

## 3.4 Comparing to independent SIT datasets

The proxy SIT product, both before and after application of the ice type-SIT correction, was compared to independent in situ and airborne SIT measurements.

### 3.4.1 ULS moorings

The Beaufort Gyre Exploration Project (BGEP) investigates basin-scale mechanisms in the Beaufort Gyre. As part of this project, the sea ice draft is measured at four locations in the Beaufort Gyre using moored upward-looking sonar (ULS) instruments (https://www2.whoi.edu/site/beaufortgyre/data/mooring-data/, Krishfield and Proshutinsky, 2006). Three of these locations fall within the area of the ice charts though only data from mooring A and D (74°59N 149°58W and 73°59N 139°59W respectively, see Figure 1) are considered as they provide a long continuous daily record of ice draft (2003-2020 for location A and 2006-2020 for location D). The sea ice draft was converted to SIT assuming hydrostatic equilibrium: $h_i = \frac{\rho_w}{\rho_i} h_d - \frac{\rho_s}{\rho_i} h_s$ , where $h_i$ is sea ice thickness, $h_d$ is sea ice draft, $h_s$ is snow depth and $\rho_w$, $\rho_i$, and $\rho_s$ are the densities of sea water, sea ice and snow respectively. Snow depth and density were retrieved from the Lagrangian snow evolution model SnowModel-LG (Liston et al., 2020; Stroeve et al., 2020), i.e. the same snow dataset used in the CryoSat-2 SIT product. The sea water density was assumed 1024 kg/m3. Sea ice density was assumed 916.7 kg/m3 when FYI and 882 kg/m3 when MYI (Alexandrov et al., 2010). The observed ULS SIT was averaged monthly and compared to the closest 10 grid cells in the SIT proxy-product.

### 3.4.2 Operation IceBridge

NASA's Operation IceBridge (OIB) provides airborne retrievals of sea ice thickness during spring using a combination of laser and radar altimeter sensors. Campaigns in April 2009, April 2010, March 2011, March 2012, March 2013 and April 2013 included flights over sea ice in the Western Canadian Arctic. SIT from the OIB L4 Sea Ice Freeboard, Snow Depth, and Thickness (IDCSI4) product (https://nsidc.org/data/idcsi4/versions/1, Kurtz et al., 2015) for these six campaigns was used. Measurements with an uncertainty higher than 1 m were removed. The spatial resolution of this product is 40 m. As we aim to compare the OIB SIT data to the SIT proxy-product we average every 1250 measurements to create a product with a spatial resolution of 50 km. When less than half of the measurements over the averaging window have no value, the sample is removed.

### 3.4.3 In situ measurements

The Canadian Ice Thickness Program has collected ice thickness and snow depth measurements on landfast ice near weather stations as far back as 1947 (https://www.canada.ca/en/environment-climate-change/services/ice-forecasts-observations/latest-conditions/archive-overview/thickness-data.html). Measurements are taken at approximately the same location every year on a weekly basis, starting after freeze-up when the ice is safe to walk on, and continuing until break-up or when the ice becomes unsafe. The data have been summarised by Brown and Cote (1992) and Howell et al. (2016). Ice thickness is measured using an auger kit or a hot wire ice thickness gauge. Three of the sites are located within the study area: Cambridge Bay, Resolute, and Eureka

(see Figure 1). This allows for validation of the SIT proxy-product in the CAA channels. We compared the fast ice thickness measurements to the closest 10 grid cells in the SIT proxy-product.

Additional observations of landfast ice thickness were collected near Eureka in March and April 2014 and April 2016 by Environment and Climate Change Canada (ECCC) (King et al., 2015, 2020), and near Cambridge Bay in May 2016, April 2017 and May 2018. Observations were collected using manual ice augers and are used for further comparison to the closest 10 grid cells in the SIT proxy-product.

## 4 Model performance

### 4.1 Model evaluation

We evaluated the model performance by first calculating the testing and training error for each of the models (each month and using both Ku-band and C-band scatterometer datasets) (Figure 5). The training error was determined from the root-mean-square error (RMSE) by testing on the same data as the model was trained on. The training error tells us how well the model captures the relation between predictor features and observations of the data it was trained on. The testing error was calculated using a 10-fold cross validation RMSE of the validation dataset, which is determined by splitting the randomly shuffled dataset into 10 groups and for each unique group to hold that group out as test data, train the model on the other 9 groups, and determining the RMSE on the test group. This is done for each of the 10 groups and then the mean of the RMSEs is taken as the testing error. The testing RMSE varies between 30 and 50 cm, depending on the month and scatterometer dataset, with the RMSE error being greater for months later in the growth season (Figure 5). The testing error is expectedly larger than the training error for each of the models. However, the difference is small ($\sim$0.05 m), which suggests the model is not over-fitted (Géron, 2019). The error is larger for the Ku-band dataset than for the C-band dataset, likely because there is more training data available with the C-band dataset as the C-band scatterometers cover the entire training period (2010-2020), whereas there is no Ku-band scatterometer available for the period 2014-2017 (see Table S3 for number of training instances), and a difference in interaction with snow and ice from the different scatterometer wavelengths (Ontstott, 1992).

We also evaluated the model by not using a randomly selected 20% of the original CryoSat-2 dataset for the training and reserving it for a validation dataset. This allowed us to plot the predicted versus observed SIT from the validation dataset (Figure 6). The trend line fits closely over the one-to-one line, showing that there is no clear over- or underestimation of the ice thickness at thin or thick ends of the scale and the Random Forest Regression captures the nonlinear relationships between the input features and SIT. However, there are outliers where the prediction is more than 1 meter larger or smaller than the observed SIT. This is likely the result of the main input data from the ice charts being polygons with homogenised fields covering large areas, so the Random Forest Regression model is incapable of predicting these small-scale local variations in thickness observed by CryoSat-2.

Finally, to analyse the model performance spatially and evaluate the model's capability to capture yearly variability, the model was trained on a dataset including all years of the CryoSat-2 record, except for the 2017-2018 winter, and used to predict the SIT for this winter (Figure 7). The predicted SIT for November 2017 very closely resembles the observed SIT. The

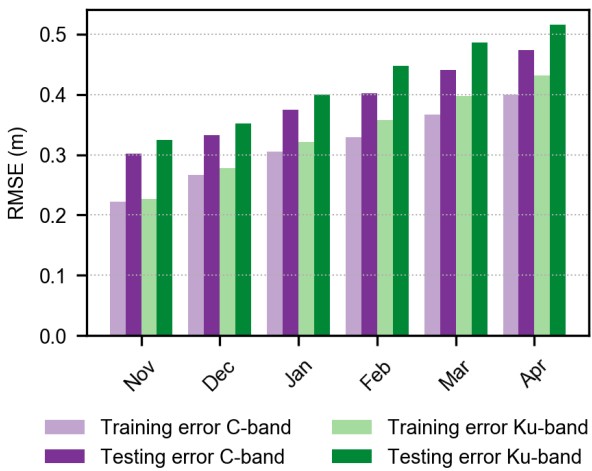

**Figure 5.** Training and testing errors in the Random Forest Regression models for each month and scatterometer frequency (C-band and Ku-band).

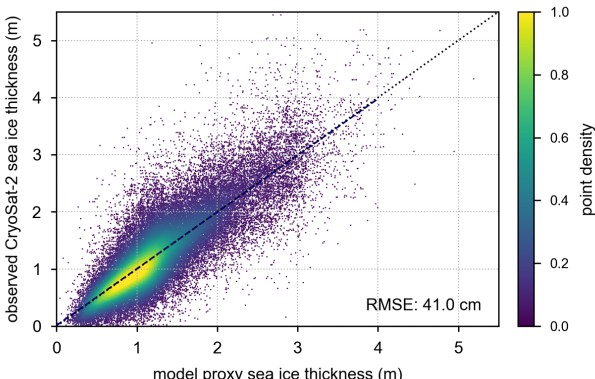

**Figure 6.** Model performance when the proxy-product is compared with CryoSat-2 observed sea ice thickness not included in the model training step. The dashed line shows the trendline closely following the dotted 1:1 line. Colours refer to point-density.

error in the prediction is larger for April 2018, although the patterns in SIT are predicted generally correctly. There is an area of the Beaufort Sea where the ice thickness is overestimated by the model and an area along the coast of Alaska where the ice thickness is underestimated compared to CryoSat-2. In these cases, the variations in CryoSat-2 SIT were not reflected by similar patterns in the ice charts or scatterometer data, and likely reflect dynamic deformation of the ice pack that underlies the ice type.

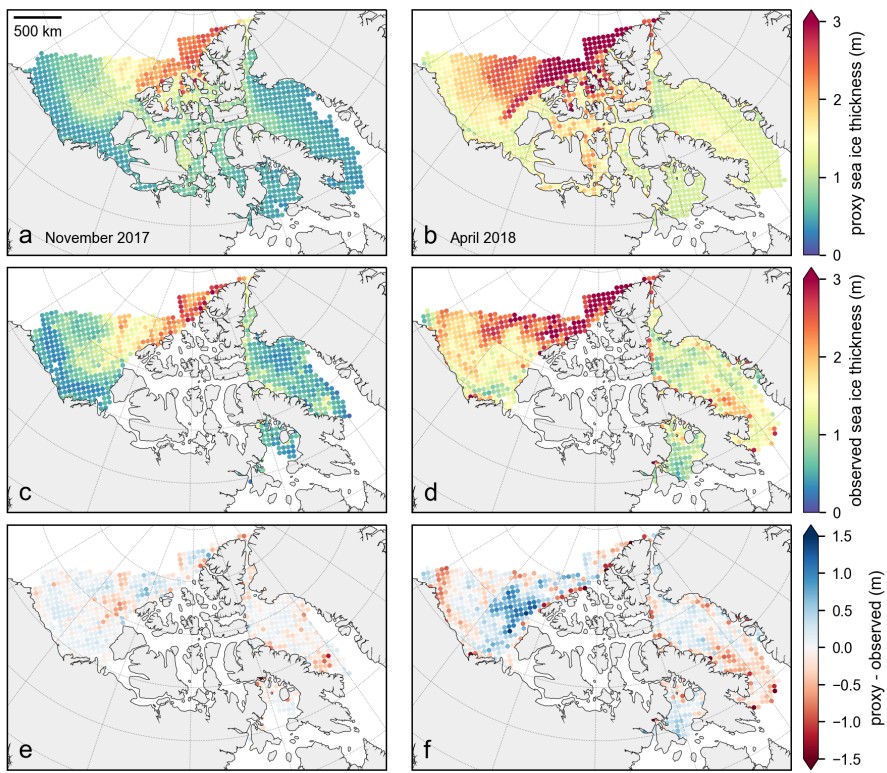

**Figure 7.** (a,b) Proxy sea ice thickness, (c,d) CryoSat-2 observed sea ice thickness, (e,f) proxy SIT-observed SIT, for (a,c,e) November 2017 and (b,d,f) April 2018, when leaving the winter 2017-2018 season of CryoSat-2 ice thickness data out of the model training dataset.

## 4.2   Comparison to independent SIT datasets

For an independent validation of the model, we compared the SIT proxy-product to in-situ and aerial observations. Typically, the validation statistics were improved with the application of the ice type-SIT correction, so we therefore use the proxy_corr SIT product for subsequent analyses.

### 4.2.1   ULS moorings

A comparison between the SIT proxy product and SIT at the BGEP ULS moorings in November and April at mooring A and D is shown in Figure 8. The mooring data allowed us to investigate the temporal performance of the proxy SIT product in the Beaufort Sea. The SIT proxy product shows similar magnitude and yearly-variability as the ULS measurements. On location A, the proxy-product and the ULS SIT have a correlation coefficient of 0.77 and an RMSE of 0.35 m, and for location D there is a correlation coefficient of 0.74 and an RMSE of 0.35 m. The RMSE is within the range of the model testing uncertainty (30-50 cm) (Figure 5), showing that the proxy product predicts SIT at this location well. Since the model testing uncertainty is

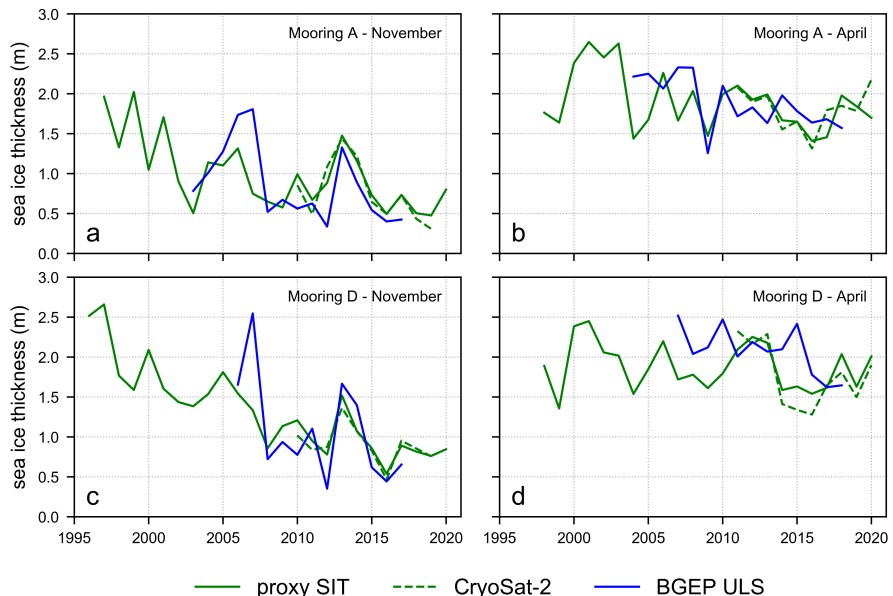

**Figure 8.** BGEP ULS moorings sea ice thickness compared to the proxy_corr sea ice thickness product and CryoSat-2 observed sea ice thickness at the same location, (a) November mooring A, (b) April mooring A, (c) November mooring D, (d) April mooring D. Locations of ULS moorings are shown in Figure 1.

only based on its performance against CryoSat-2 SIT observations, which have their own uncertainties, it is encouraging that there is a similar RMSE when comparing the SIT proxy product to independent ULS observations.

We determined the anomaly correlation coefficient (ACC) as 0.45 and 0.49 at location A and D, respectively, after removing the climatological seasonal cycle from both datasets. The ACCs are improved by 26% on average when using the proxy_corr, rather than proxy_nocorr, SIT product. These positive ACCs show that the yearly variability between the proxy-product and the ULS SIT is comparable and typically going in the same direction. The model is thus capable of determining an anomalously high or low SIT year at these two locations; however, the magnitudes of the ice thickness anomalies can be up to 0.5 to 1 m different. As an example, the proxy product correctly estimates the high SIT in November 2013 and the low SIT in November 2016 at both mooring locations in the Beaufort Gyre (Fig 7). The proxy product correctly estimates the SIT in November 2012 to be below average at mooring D (at 0.78 m), but does not get the magnitude of the SIT minimum right (0.34 m according to the mooring), and it does not estimate November 2012 to be below average at mooring A. However, the proxy product does estimate SIT very close to the observed CryoSat-2 SIT on the location of the ULS BGEP moorings for November, and as the model is trained on CryoSat-2 data it is not expected to do better than CryoSat-2. Moreover, the mean seasonal cycle is captured very well, with a mean difference between the proxy SIT and the ULS mooring of only 1 cm in November and 14 cm in April (see Figure S2 in Supplementary Materials).

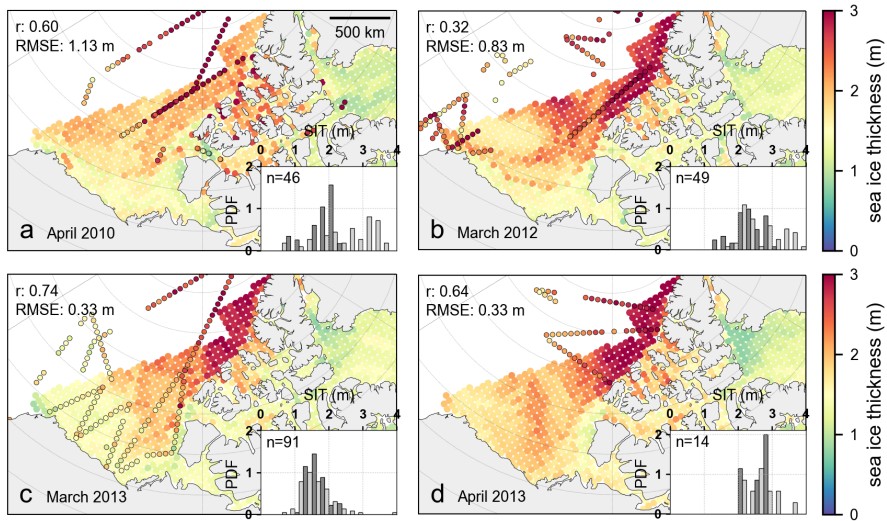

**Figure 9.** Operation IceBridge sea ice thickness measurements compare to the proxy_corr sea ice thickness product for (a) April 2010, (b) March 2012, (c) March 2013, (d) April 2013. Subfigure shows the probability density function for OIB (lightgrey) and the proxy SIT product (darkgrey) on the OIB tracks.

### 4.2.2  Operation IceBridge

The results of the comparison of the SIT proxy-product with overlapping Operation IceBridge tracks gives us insight into how well the proxy product captures regional variability in the Beaufort Sea. The proxy-product generally captures the spatial patterns and magnitudes of SIT as observed by OIB campaigns during the end of winter (Figure 9). The SIT proxy-product correctly captures the spatial pattern of thicker sea ice in the north-eastern Beaufort Sea and thinner sea ice to the south.

The proxy-product is found to underestimate ice thickness along the west coast of the CAA, north of the Queen Elizabeth

Islands, in April 2010 and along the north coast of Alaska near Point Barrow in March 2012. Both these months are characterised by high RMSEs when comparing OIB with the proxy SIT product (1.13 m in April 2010 and 0.83 m in March 2012). In April 2010 the input data in the model is characterised by smaller floe sizes in the ice chart than other years. Upon manual investigation of radar imagery of this region, the floe size does not seem smaller than in other years. This difference in floe size caused the model to predict thinner ice in April 2010 than in other years and likely lead to the underestimation. This shows that

the SIT proxy-product created by this model is reliant on the consistency of the manually created ice charts which, although generally robust (Tivy et al., 2011), can include anomalies.

In March 2012, the area of thick ice near the coast of Point Barrow, Alaska, observed by OIB was also observed within the CryoSat-2 SIT product but was underestimated by the proxy-product. This thicker region in the OIB measurements is potentially caused by dynamic thickening of the ice pack as it converges against the coast (Fukamachi et al., 2017; Babb

et al., 2020). Given that ridged ice is not classified in the ice charts, the influence of this process would not be captured by the

machine-learning model, which in turn highlights one of the limitations. A similar phenomenon might be visible in April 2018 (Figure 7), where the SIT proxy-product shows thinner results than the CryoSat-2 observations in this region. For comparison ice thickness in this region was more accurately predicted by the proxy-product in March 2013 (Figure 9c), which likely means that there was less dynamic thickening in 2013 than in 2012. Dynamic thickening of FYI during the growth season might also
be the cause of the model testing error being higher in the months at the end of winter.

March and April 2013 (Figure 9c and 9d) are characterised by higher correlations and lower RMSEs (0.33 m for both March and April 2013) when comparing the proxy SIT product to OIB.

### 4.2.3 In situ measurements

Landfast SIT measurements at weather stations in the CAA provide a comparison to the entire record of the SIT proxy-product
in the Canadian Arctic channels (Figure 10). A comparison between the fast ice thickness record and the proxy-product in Cambridge Bay and Resolute give high correlations of 0.93 and 0.73 respectively, and low RMSEs of 0.19 m and 0.26 m respectively. The weather station at Eureka provides a lower correlation (0.62) and higher RMSE (of 0.36 m). However, the RMSEs of all three locations are within the uncertainty of the SIT proxy-product of 0.3 to 0.5 m. The reliability of the model testing uncertainty (Figure 5) is reinforced by the close comparisons to all three independent validation exercises here. The
range of in-situ field measurements in Eureka and in Cambridge Bay agree with both the fast ice weather station measurements and the proxy SIT product (Figure 10).

The proxy SIT product overestimates sea ice thickness in the start of winter (November and December) and underestimates sea ice thickness at the end of winter (February, March, April) for all three locations (Figure 10). This shows that the seasonal cycle in sea ice growth is not fully captured by the proxy SIT product in the CAA. However, it also needs to be noted that
the observation at the weather stations are made in fast ice in easily accessible locations very close to shore, and may not be representative of the general regional ice conditions.

The anomaly correlation coefficient between landfast ice measurements and the proxy product showed that the interannual variability was well captured in Cambridge Bay (ACC of 0.37) and moderately well in Eureka and Resolute (0.11 and 0.20, respectively). Again, these positive ACCs demonstrate that the directions of the interannual variations in SIT anomalies (higher
or lower than usual) are typically the same between the in-situ data and the proxy product. However, the magnitudes of the anomalies can be different. The ACCs are improved by 42% on average when using the ice type-corrected, rather than uncorrected, SIT proxy product.

The correlation and anomaly correlation between the in-situ measurements and the SIT proxy-product at the landfast ice weather stations is in the same range as at the ULS moorings in the Beaufort Gyre. This indicates that the proxy product can
estimate SIT in the channels of the Canadian Arctic Archipelago as accurately as in the open area of the Beaufort Sea.

### 4.3 Limitations and potential

As the Random Forest Regression model is trained on CryoSat-2 sea ice thickness observations, the results can only be as good as CryoSat-2 observations. This is illustrated well in Figure 8, where the SIT proxy product does in places differ from the local

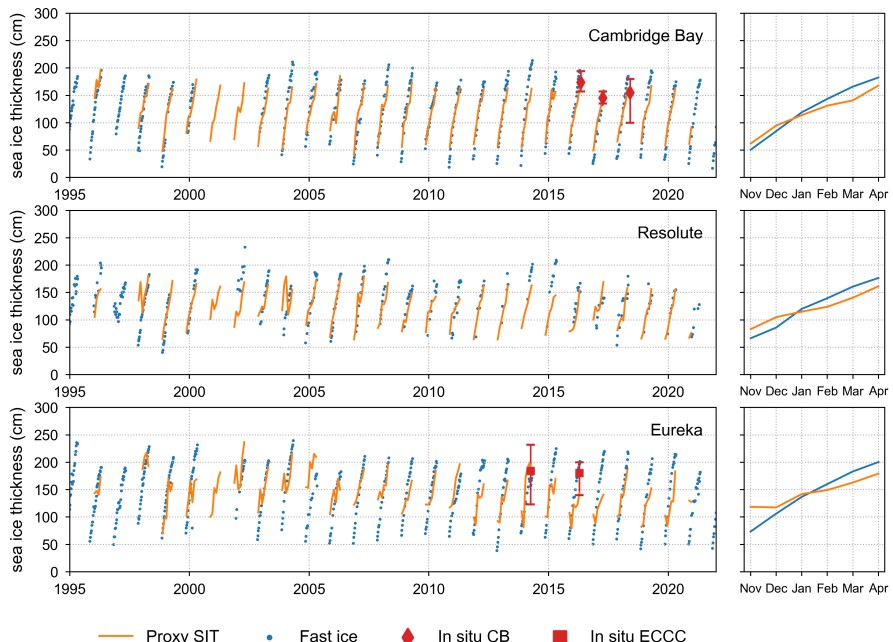

**Figure 10.** Fast ice thickness measurements (for locations see Fig 1), proxy_corr sea ice thickness product, and field campaign in-situ observations from ECCC in Eureka (King et al., 2015, 2020) and Cambridge Bay. Graphs on right-hand side show mean seasonal cycle for each location.

ULS SIT observations, but agrees well with CryoSat-2 observations in all panels. There are known limitations to CryoSat-2
SIT retrievals – e.g., the likely incorrect assumption that the Ku-band radar signal penetrates the snow layer in all cases (Willatt et al., 2011; Nandan et al., 2017; Stroeve et al., 2022; Nab et al., 2023), the instrument not being able to measure freeboards lower than 2.5 cm (Landy et al., 2020), and the need for a reliable snow depth product to convert from radar freeboard to SIT (Glissenaar et al., 2021) – which will propagate into this SIT proxy-product.

Another limitation of the proxy product is its reliance on reliable and consistent ice charts, which are created manually by
ice analysts from different data sources. The data sources available, and thus the quality of the ice charts, change over time, with a big increase in quality of the ice charts in 1996 with the introduction of RADARSAT satellite observations (Tivy et al., 2011), which is why we select 1996 as the start year of the proxy SIT product. Tivy et al. (2011) have assessed the data quality of the ice charts and determined that all regions have a high enough quality since 1996 for any statistical analysis. Nevertheless, there is some variability in the quality over time and per region, with the quality being higher in the Beaufort Sea, Baffin Bay,
and Parry Channel, and slightly lower in the Arctic Ocean Periphery (Tivy et al., 2011).

The proxy SIT product has difficulty resolving sea ice thickness in heavily ridged regions, as the ice charts do not specify ridging. This is particularly true in MYI regions, as there are no sub-categories for thin or thick MYI, and the scatterometer

backscatter shows no difference between thin and thick MYI. Because of this the SIT proxy struggles to capture SIT in regions with a lot of ridged MYI.

Without applying the ice type-SIT correction to the proxy product, the proxy_nocorr SIT assumes that the relation between the model features (ice type, form of ice, and scatterometer backscatter) and sea ice thickness stays constant over time. This is why the ice type-SIT correction is applied to create the proxy_corr SIT product. The proxy_corr SIT product showed better agreement with the independent SIT datasets used for validation than the proxy_nocorr SIT product, showing that applying this correction improves the results. However, the ice type-SIT correction is determined using PIOMAS, which is a sea ice model with known limitations and uncertainties. One of these limitations is that PIOMAS is known to overestimate thin ice thickness and underestimate thick ice thickness (Schweiger et al., 2011), underestimating negative trends compared to observations. For this reason the applied ice type-SIT correction might be underestimated in this study, leading to conservative trend estimations.

A potential of the presented method in retrieving the proxy sea ice thickness product is that this method can be applied to new ice charts and scatterometer data as they are released. A version of the model based solely on weekly ice charts that does not include scatterometer data, which has a delayed release, offers the potential for near-real time estimates of sea ice thickness, though this comes with an associated 2-10 cm increase in the error.

Additionally, the new year-round sea ice thickness record from CryoSat-2 (Landy et al., 2022) creates the potential to extend this methodology to extend the proxy of SIT into the summer months. This does however come with its own separate challenges, including a lower amount of training data due to less sea ice and a change in scatterometer backscatter with snow melt, and is therefore not considered in this analysis.

## 5 Sea ice thickness proxy-product (1996-2020)

Using a combination of remotely sensed sea ice products we have created a proxy sea ice thickness record that covers the full Canadian Arctic, including the CAA, and extends back to 1996. We present a proxy_nocorr SIT product, which can be used to study sea ice thickness trends caused by changes in ice type, and a proxy_corr SIT product, which is corrected for ice type-SIT trends and can be used to research long-term sea ice thickness trends. We focus our discussion on the proxy_corr SIT product, as this showed better statistics in the validation with the independent SIT datasets. The proxy_corr product compares well with in-situ observations and captures the general spatial pattern of thicker sea ice in areas known to contain old ice (the north-eastern Beaufort Sea and the northern channels in the CAA), and thinner ice in areas that are typically ice-free during summer (i.e., Southern Beaufort Sea, Foxe Basin, Baffin Bay) (Figure 11). The product also highlights a general reduction in sea ice thickness over the 25-year study period (Figure 11), though there is a considerable variability in the trends both spatially and temporally (Table 3).

The overall trends in sea ice thickness in the region show significant thinning throughout winter (Table 3). Thinning is strongest in early winter (November-January) and less pronounced in the later winter months (February-April). This indicates a later freeze-up in recent years with thickening of the sea ice happening later in winter.

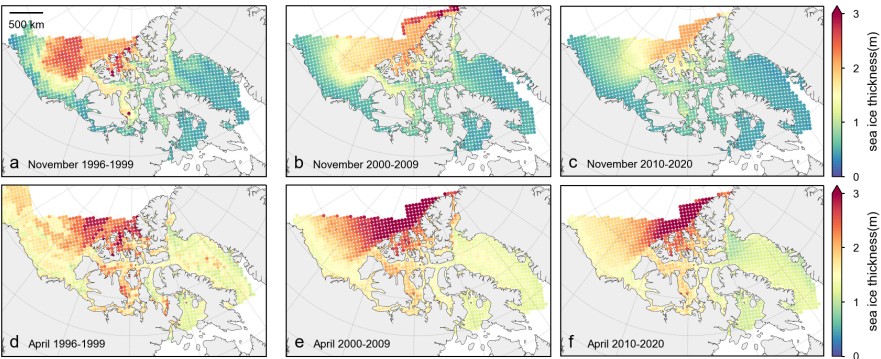

**Figure 11.** Decadal means from the proxy_corr sea ice thickness product for (a,b,c) November and (d,e,f) April (a,d) 1992-1999, (b,e) 2000-2009, and (c,f) 2010-2020.

The interannual variability in the proxy product is characterised by the residual standard error (RSE), which characterises standard deviation of the residuals in a regression model and thus the variability from the trend:

$$RSE = \sqrt{\frac{\sum (y_i - \hat{y}_i)^2}{n-2}} \qquad (2)$$

where $y_i$ is the observed value of mean SIT in the proxy product for a given year, $\hat{y}_i$ is the expected value in the fitted linear regression model for the same year, and $n$ is the number of years. In the study area, the interannual variability is largest in December, with an RSE of 13 cm. The smallest interannual variability is found in April, with an RSE of 6 cm. Of the four regions outlined in Figure 12, the variability is largest in the Arctic Ocean Periphery, with a maximum RSE (interannual variability) in January of 46 cm. High interannual variability is also found in the channels of the Queen Elizabeth Islands in the north part of the archipelago. The lowest interannual variability is found in Baffin Bay, with a maximum RSE in February of 12 cm.

Regionally, April SIT trends are largest in Baffin Bay and Arctic Ocean Periphery (Figure 12), and November SIT trends are largest in the Beaufort Sea and Arctic Ocean Periphery (Figure 13). Trends in the CAA are variable during April but show a relatively coherent reduction in ice thickness during November (Figure 12 and 13).

Sea ice in northern Baffin Bay shows significant (p<0.05) thinning of locally up to 30 cm/decade in April for the full time period (1996-2020). Sea ice thickness trends in Baffin Bay have been difficult to determine in the past because altimetry records are highly reliant on the selected snow depth record and processing methods (Glissenaar et al., 2021). All altimetry records in spring show thinning in the North Water Polynya region in the north of Baffin Bay over the past 20 years (Glissenaar et al., 2021), agreeing with the presented record here. More uncertain are sea ice thickness trends in the southern part of Baffin Bay, where trends in altimetry records are highly variable and dependent on the snow depth product applied (Glissenaar et al., 2021). The proxy SIT product presented here shows an asymmetric SIT trend in Baffin Bay, with no change in the southwest and thinning in the north and northeast. This is mostly caused by a decrease in the ice type 'thick FYI' and an increase in the

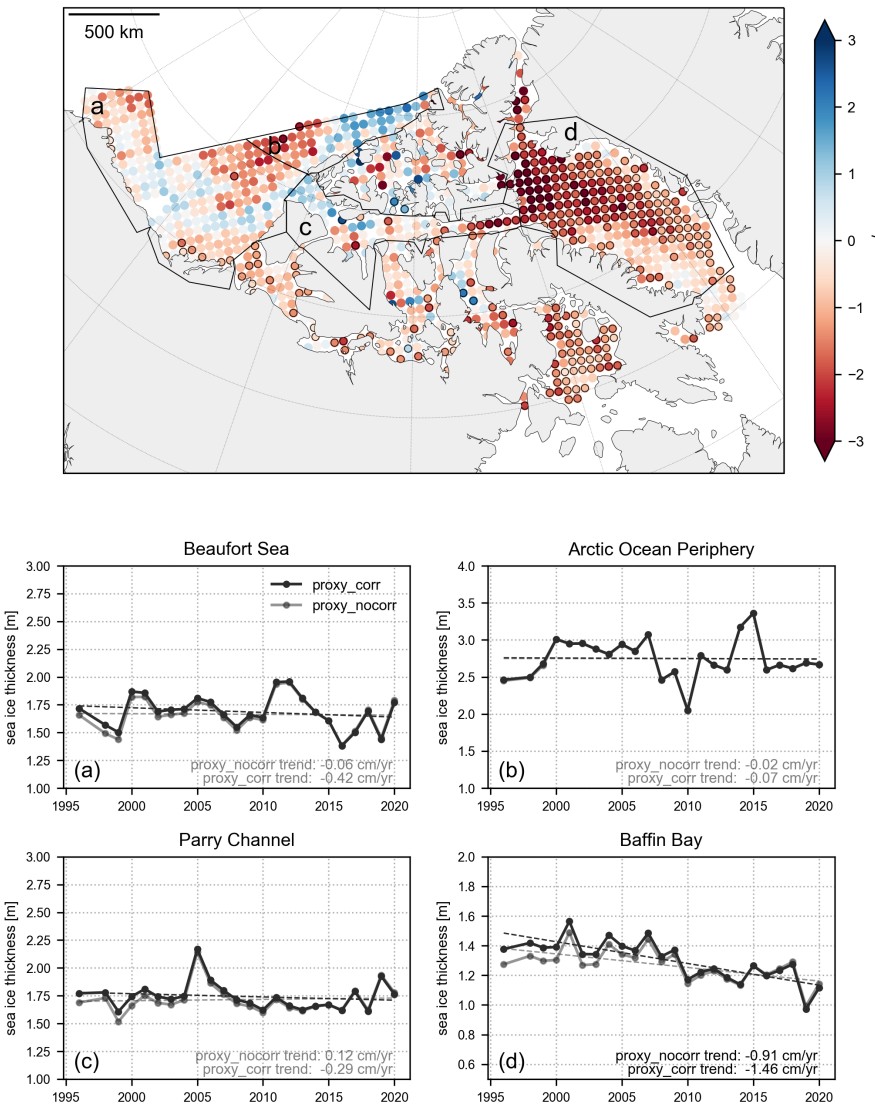

**Figure 12.** Sea ice thickness trends from the proxy product for April 1996-2020. The map shows trends from the proxy_corr SIT product, solid outline circles where statistically significant (p<0.05). Timelines given for the mean of subregions of the Canadian Arctic (a,b,c,d). Trends shown for the proxy_nocorr SIT product (light) and the proxy_corr SIT product (dark). Trend numbers are bold where significant (p>0.05).

ice type 'medium FYI' in the east and north of Baffin Bay. There is no significant change in the MYI concentration in Baffin Bay in April.

Sea ice in part of the Arctic Ocean Periphery north of the Queen Elizabeth Islands shows non-significant ice thickening in April. This thickening coincides with a non-significant increase in scatterometer backscatter and a significant but small

replacement of FYI by MYI in the ice charts, so thickening could be caused by more MYI convergence against the coast (Kwok, 2015).

The Parry Channel, an important area for shipping activities, shows large yearly variability but no significant change in April (Figure 12c). The interannual variability in the Parry Channel is linked to the variability and trends in MYI in this region. The time series of MYI in this region as shown by Howell et al. (2022) is very similar to the time series in the proxy product for

SIT (Figure 12), with a correlation between the two of r=0.64. This implies that in a heavy MYI year, the mean ice thickness in Parry Channel is around 2 m, whereas in a low MYI year the mean ice thickness is around 1.5 m. This highlights the importance of MYI advection within the CAA and its role in conditioning the ice cover for the melt season and shipping season.

The end of winter (April) SIT shows non-significant thinning in the Beaufort Sea caused by a decline in the old ice concentration in the Beaufort Sea over the study period, as indicated in the old ice category of the ice charts and the scatterometer

backscatter. This decline in old ice in the Beaufort Sea is likely caused by an increase in MYI melt in the Beaufort Sea itself, as the influx of MYI from the north has been shown to have increased (Babb et al., 2022).

Seasonally, the trends also vary by region. Baffin Bay and the Beaufort Sea have significant negative trends in SIT for almost every month in the study period in the proxy_corr product (Table 3). In the Beaufort Sea thinning is most pronounced at the start of the growth season (-28 cm/decade in November; Table 3), which is associated with a stronger negative trend in old ice

in November than April caused by greater reductions in old ice surviving the summer but a continued replenishment of old ice from the Arctic Ocean Periphery in winter (Babb et al., 2022). In Baffin Bay the thinning is more pronounced in spring (-15 cm/decade in March and April; Table 3), mostly because of strong thinning in the north of Baffin Bay, which is where the Northwater Polynya is located and corresponds to a more active polynya and greater occurrence of thin ice since the 1990s (Preußer et al., 2015). In Parry Channel thinning is only statistically significant in autumn (-18 cm/decade in November, -20

cm/decade in December), with more variability in mid- and late-winter. This is relevant for shipping safety as thinning of the sea ice in autumn would lengthen the summer shipping season (Howell et al., 2022; Mudryk et al., 2021). A comparison of the proxy SIT product with the Alfred Wegener Institute (AWI) CryoSat-2 SIT product (Hendricks and Paul, 2022) in the channels of the CAA (Figure S3 in Supplementary Materials) showed a good agreement in November and a much better spatial coverage by the proxy SIT product in April. The Arctic Ocean Periphery shows no significant thinning throughout winter.

## 405   6  Conclusions

We present a proxy sea ice thickness product for the Canadian Arctic for 1996-2020 based on long-term remote sensing records. The presented sea-ice thickness proxy-product estimates sea ice thickness with 30 to 50 cm testing uncertainty, verified in a comparison with independent ice draft and thickness observations. The proxy-product for SIT goes further back in time than satellite altimetry records, offering the opportunity to study trends and variability in SIT on longer timescales, and offers

complete coverage of the Canadian Arctic, including coastal areas and the CAA where the use of altimetry to estimate ice thickness is less certain. The presented proxy-product is the first large-scale SIT product reliably covering the complex CAA channels.

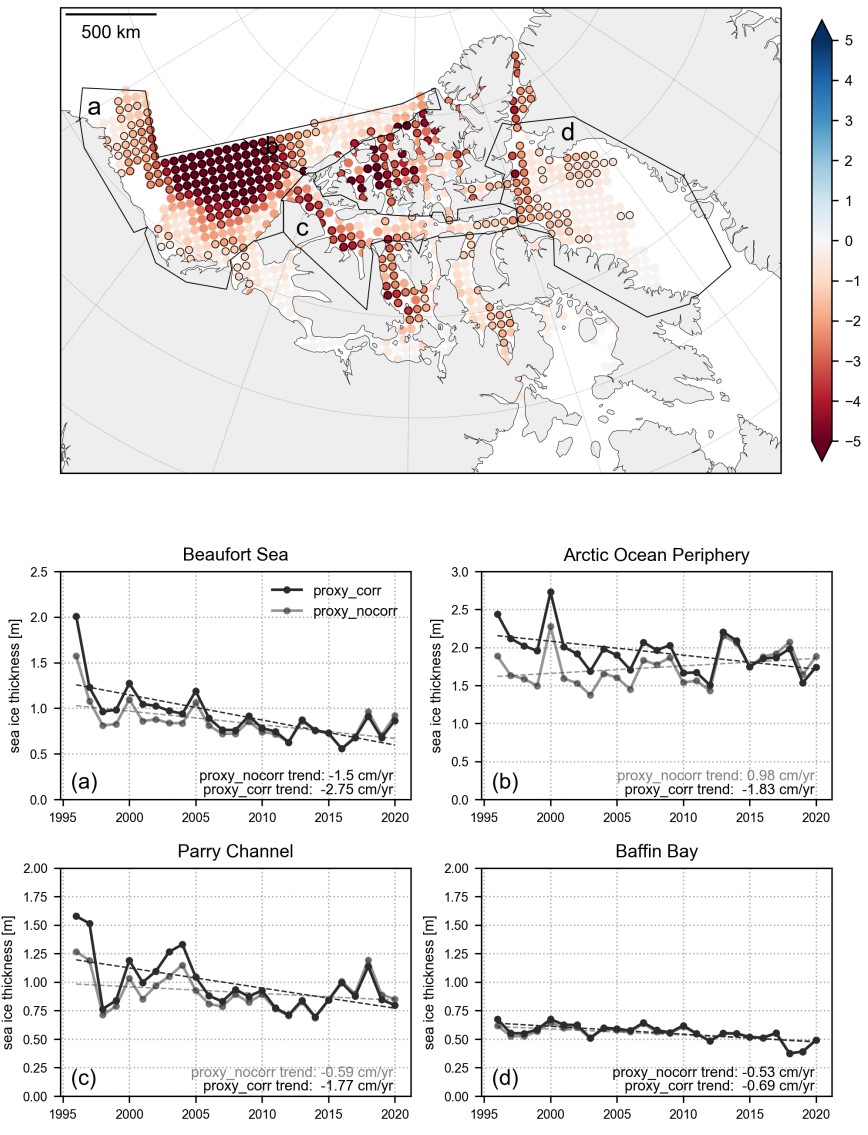

**Figure 13.** Same as Figure 12 but for November.

The sea ice thickness proxy product shows that sea ice is thinning over most of the Canadian Arctic, with a mean trend over the full area of -0.82 cm/yr in April. The trends vary locally and throughout winter. The Beaufort Sea and Baffin Bay show significant thinning during all months, while the Arctic Ocean Periphery shows negative trends during all months but January and March, and Parry Channel shows negative trends during November and December. Thinning in the Beaufort Sea peaks at -2.8 cm/yr in autumn (November), whereas Baffin Bay shows the strongest thinning in spring (-1.5 cm/yr in April). The

**Table 3.** Sea ice thickness trends for proxy_nocorr and proxy_corr in Canadian Arctic subregions 1996-2020 in cm/yr. Subregions outlines shown in Figure 12/13. Bold where significant ($p<0.05$).

|  |  | Beaufort Sea | Arctic Ocean Periphery | Parry Channel | Baffin Bay | Full study area |
|---|---|---|---|---|---|---|
| November | proxy_nocorr | **-1.50** | 0.98 | -0.59 | **-0.53** | **-0.67** |
|  | proxy_corr | **-2.75** | **-1.83** | **-1.77** | **-0.69** | **-1.54** |
| December | proxy_nocorr | -0.34 | -0.29 | -0.16 | -0.26 | -0.57 |
|  | proxy_corr | **-2.11** | **-3.87** | **-2.04** | **-0.93** | **-2.11** |
| January | proxy_nocorr | **-1.18** | 0.54 | 0.61 | -0.47 | -0.47 |
|  | proxy_corr | **-2.54** | -2.05 | -0.57 | **-0.96** | **-1.48** |
| February | proxy_nocorr | **-0.77** | -0.57 | 0.26 | -0.65 | -0.44 |
|  | proxy_corr | **-1.85** | **-2.26** | -0.70 | **-1.30** | **-1.34** |
| March | proxy_nocorr | **-0.86** | 0.44 | 0.27 | **-1.00** | **-0.57** |
|  | proxy_corr | **-1.75** | -0.98 | -0.47 | **-1.54** | **-1.31** |
| April | proxy_nocorr | -0.06 | 0.02 | 0.12 | **-0.91** | -0.38 |
|  | proxy_corr | -0.42 | -0.07 | -0.29 | **-1.46** | **-0.82** |

Arctic Ocean Periphery shows the highest interannual variability. Thinning in Parry Channel peaks during autumn (-2.0 cm/yr in December).

The SIT proxy-product can be used to study long-term trends and variability in SIT in the Canadian Arctic, to monitor SIT for shipping safety purposes, and for the initialisation and verification of seasonal prediction models. The product can also be used as reference or in models for studying other features in this area that are affected by SIT change. For example, research towards primary productivity and microbial life (Post et al., 2013; Campbell et al., 2022), the effect of oil pollution (Redmond Roche and King, 2022), and the surface energy balance (Ledley, 1988). Lastly, the Random Forest Regression can be applied in near-real time to estimate ice thickness from ice charts and scatterometer data and extend the proxy SIT product into the future.

*Code and data availability.* The data and code used to create and analyse the dataset are available on https://doi.org/10.5281/zenodo.7644053. The created sea ice thickness proxy dataset is also separately available on https://doi.org/10.5281/zenodo.7644085.

## Appendix A: Scatterometer record

One of the features used in the Random Forest Regression model is scatterometer data. As there is no continuous record of one instrument over the entire 1996-2020 record, we use data of multiple instruments. We've used data from both C-band and Ku-band scatterometer instruments. C-band scatterometers work in the 4-8 GHz frequency range and Ku-band in the 12-18 GHz frequency range. As these wavelengths interact differently with snow and ice (Ontstott, 1992), we expect C-band and Ku-band instruments to give different results and do not combine scatterometer data from the different bands into one record.

Instead, we create a C-band record combining ERS-1, ERS-2, and ASCAT, and a Ku-band record combining QuickSCAT, OSCAT-1, and OSCAT-2 (Figure 3). This section discusses why we believe these records can be combined.

  NASA's QuickScat (Quick Scatterometer) was an Earth observation satellite carrying the Ku-band (13.4 GHz) dual polarization scatterometer SeaWinds. QuickScat was launched on 19 June 1999 and stopped collecting data on 21 November 2009. Daily horizontal polarization gridded data was retrieved from NASA SCP for 1 July 1999 to 21 November 2009

(https://www.scp.byu.edu/data/Quikscat/SIRv2/Quikscat_sirV2.html).

  The Indian Space Research Organisation's (ISRO) OSCAT-1 scatterometer was carried by Oceansat-2 and operated in Ku-band (13.515 GHz). The instrument provides daily global coverage at a resolution of 25 km. Horizontal polarization gridded scatterometer data was retrieved from NASA SCP for 5 November 2009 to 21 February 2014 (https://www.scp.byu.edu/data/OSCAT/SIR/OSCAT_sir.html). The follow-on mission ScatSat-1 carried OSCAT-2. Daily horizontal polarization data was

obtained from MOSDAC for 1 November 2016 to 31 December 2020 (https://mosdac.gov.in/satellite-catalog).

  The Ku-band record consists of QuickScat (data from NASA SCP available for August 1999 to 23 November 2009), OSCAT-1 (data available 5 November 2009 to 21 February 2014), and OSCAT-2 (data from MOSDAC available for November 2016 to present). OSCAT-1 and OSCAT-2 are similar instruments. The NASA SCP OSCAT-2 dataset is only available until 2019, so we decide to use the MOSDAC OSCAT-2 dataset. OSCAT-1 and OSCAT-2 do not have a temporal overlap, so a direct comparison

is not possible. Figure A1 shows that the seasonal cycle of the retrieved backscatter signal is very comparable between the two products, so we do not apply a bias correction and assume the records can be combined. OSCAT and QuickScat are very similar instruments. OSCAT measurements are at a slightly different incidence angle than QuickScat. OSCAT-1 and QuickScat have a 19-day temporal overlap in November 2009. The backscatter from both instruments in this period are compared (Figure A2) and deemed similar enough to combine the record.

ERS-1 was a European Space Agency (ESA) spacecraft launched on 17 July 1991 to provide microwave spectrum-based environmental monitoring. The spacecraft carried a range of instruments, including the Wind Scatterometer and Synthetic Aperture Radar (SAR) instruments, which worked in tandem in a configuration called the Active Microwave Instrument (AMI). The instrument measures C-band in a frequency of 5.3 GHz and has a spatial resolution of about 50 km. The ERS-1 mission ended on 10 March 2000. Gridded ERS-1 scatterometer data was retrieved from the NASA Scatterometer Climate Record Pathfinder

(SCP) for 1 January 1996 to 2 May 1996 (https://www.scp.byu.edu/data/ERS/SIR/ERS_sir.html). ERS-2 was launched in the same orbit as ERS-1 on 21 April 1995 and carried the same instruments as ERS-1. ERS-2 was taken out of service on 5

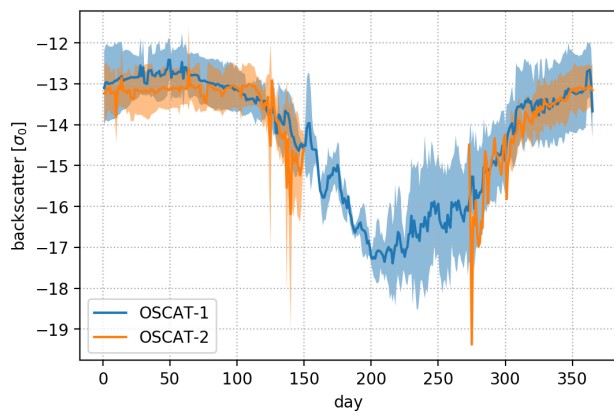

**Figure A1.** Seasonal cycle of scatterometer backscatter from OSCAT-1 and OSCAT-2.

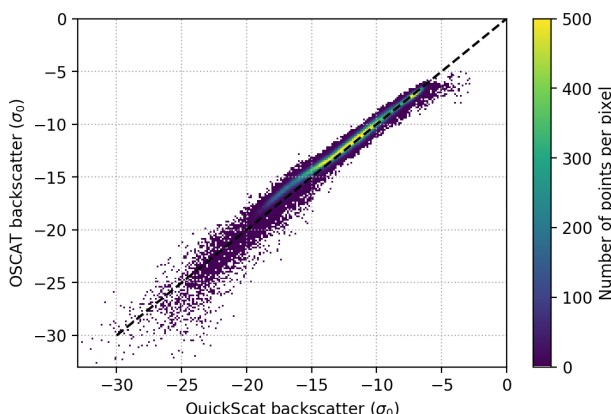

**Figure A2.** Backscatter from OSCAT-1 and QuickScat for the overlapping period (5-23 November 2009).

September 2011. Gridded ERS-2 scatterometer data for 1 June 1996 to 18 January 2001 was also retrieved from NASA SCP (https://www.scp.byu.edu/data/ERS/SIR/ERS_sir.html).

The Advanced Scatterometer ASCAT is a C-band (5.255 GHz) advanced version of the AMI instrument flown on ERS-1 and
ERS-2. ASCAT was carried by ESA's Meteorological operational satellite A (MetopA), which was part of the EUMETSAT Polar System, and Metop-B and -C. Metop-A was launched on 19 October 2006 and retired on 15 November 2021. Metop-B and -C were launched in 2012 and 2018 respectively and are still operational. It provides global coverage in 1.5 days and has a 12.5 km spatial resolution. ASCAT's vertical polarization scatterometer gridded data was retrieved from NASA SCP for 1 January 2007 to 31 December 2020 (https://www.scp.byu.edu/data/Ascat/SIR/Ascat_sir.html).
The C-band record consists of ERS-1 (data from NASA SCP available for January 1992 to April 1996), ERS-2 (data available June 1996 to mid-January 2001), and ASCAT (January 2007 to present). ERS-1 and ERS-2 are designed to be

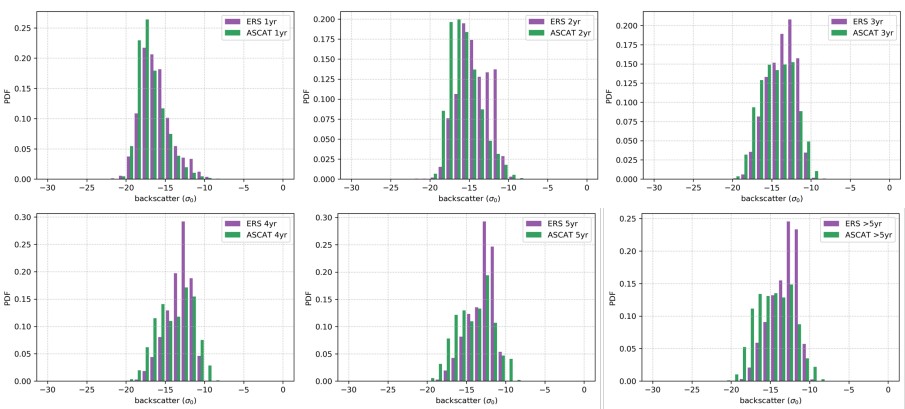

**Figure A3.** Probability density function of backscatter on different ice ages for ERS-2 and ASCAT.

identical twins, with the same scatterometer instrument and flying in the same orbit. They can thus be applied together. AS-CAT differs from ERS by its higher observation density, better noise characteristics, and slightly higher incidence angles. As we use a low-resolution product for the scatterometer data (GRD), we believe the difference in noise is removed. The lower observation density of ERS is no issue because only monthly data is used. There is no temporal overlap in ERS-2 and ASCAT, so a direct assessment of their comparison is not possible. As the period between ERS-2 and ASCAT has seen a decline in older ice, we do not expect the scatterometer data to be directly comparable. We have compared the backscatter results for ERS-2 and ASCAT for different ice ages (obtained from the EASE-grid Sea Ice Age product from NSIDC, https://nsidc.org/data/nsidc0611/versions/4), and show that the backscatter distribution for ice of the same age is similar for ERS2 and ASCAT (Figure A3).

*Author contributions.* IAG conceptualised the study, carried out the main analysis and wrote the paper. JCL provided the SIT from CS2 and helped develop the methodology. JCL, DGB, GJD, and SELH contributed to the interpretation of the results. All authors contributed to revising and improving the manuscript.

*Competing interests.* The authors declare no competing interests.

*Acknowledgements.* We would like to acknowledge Vishnu Nandan and John Yackel for supplying sea ice thickness observations from Cambridge Bay May 2016, April 2017, May 2018. This work was funded primarily by an internal University of Bristol PGR Scholarship of I. Glissenaar. J. Landy was supported by the INTERAAC project under grant 328957 from the Research Council of Norway and by the SUDARCO project under grant 2551323 from the High North Research Centre for Climate and the Environment (The Fram Centre). D.

Babb is supported by the Natural Sciences and Engineering Research Council of Canada (NSERC) as well as the Canada Research Chair (CRC – D. Barber) and Canada Excellence Research Chair (CERC – D. Dahl-Jensen) programs.

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
