# Peer review of "A long-term proxy for sea ice thickness in the Canadian Arctic: 1996-2020"

_EGUsphere, 2023_

## Referee Comment (RC1)

A long-term proxy for sea ice thickness int the Canadian Arctic: 1996-2020

Authors present for the first time a long time terme record (25 years), in the Canadian archipelago, for winter sea ice thickness. The thickness proxy is estimated using a neural network approach trained on CS-2, using Ice charts and backscatter information from scatterometers. Thickness are evaluated with in situ measurements within the Beaufort sea and coastal stations as well as with OIB airborne campaigns. This time series is the first one presented for Canadian Arctic, applications of the time series are quite large and the method can be applied for NRT purpose. The study also provides a first overview of sea ice thickness evolution within the Canadian arctic from observation during the past 25 years, where altimetry generally fails at estimating sea ice thickness, especially for long term records.

(I tried to read the data, but the dataset is not as easy readable as I expected, it is not gridded.)

General comment :

The overall study is well presented, clear and the validation convincing. The product is well discussed, especially in the limitation section with CS-2, and limitation of Ice charts for this type of study.
I have two main concerns, concerning two points of the methodology.

The first one, relatively minor, concerning the representativity of the dataset, to what extent are the data used to train the NN over the period with CS-2 representative of the whole area you are trying to estimate, in other words, is there a bias in the sampling of the dataset that can lead to a bias in the predicted SIT. For instance, is the backscatter of CS-2 'area' is representative of CS-2+channels area ? A simple plot of the backscatter distribution in the region used for training and the global region would give a good idea, same for the parameters used from ice charts that allow to train the NN (floe width etc)

The second one concerns the data « trending » that is made for MYI thinning correction. To what extent PIOMAS trend for MYI SIT is supposed to represent the real trend especially before 2010 as we usually see higher discrepancies while going back in time with PIOMAS ? This can maybe be discussed a bit more in the conclusion or in a discussion section ? Especially that, as explained by the authors, PIOMAS thick ice (MYI ?) is underestimated, linearly for the past 25 years ?
How confident are the authors by « retrending » using PIOMAS ?
I was also wondering it which extent does it make sens to trend the data and then to explain changes in the trend with the data as we could directly study PIOMAS dataset to get these trends ?
I am a bit confused with the corrected and not corrected trends products. Why the authors detail trends and explain changes for the not corrected sea ice thickness product, as it is supposed not to reflect the trends ? Especially regarding the differences between not corrected and corrected trends for all regions and all months.
Maybe in a way the study of the trends with the corrected dataset will be more relevant if it would have been validated before as the SIT changed, but I am not sure that is the purpose of this study. What bothers me is probably L 170, and that SIT are consistent but not to estimate trends… so the dataset is not so consistent as correcting the SIT by the trend will change the SIT values.
Same for fig 11 and 12, the not corrected trend values are shown but they are supposed not to be so relevant. Could you make Fig 11/12 (maps) with the trends for corrected SIT ? Are the spatial patterns similar ? (Not necessarily in the manuscript for now.)

I considered major revision because it can lead to some changes in the manuscript but the overall methodology and the manuscript as well as the validation part and the discussion part is very relevant. I am just wondering if the dataset you are validating is the good one. Which dataset would you advise me to use for any application ? I think that this is the one you should validate and I feel the way it is presented a bit confusing.

Specific comments :

L 2/3 : You should be mentioned in the abstract that the estimation is based on Ice chart / scatterometers as it is one of main elements of the thickness estimation.

L 6: 'mean trend' is a bit confusing. If I am not mistaking, this is not the mean trend this is the trend for the non corrected product for all the period and the whole area.

L128 : why to chose the mean of the two product as you have seen that ku seems to be more more uncertain (higher RMSE) ?

L 176 : Just to be sure, are you trending the SIT as following :
$$(SIT_{corr_m} = SIT \cdot t \cdot (trend_{MYI_m} \cdot C_{MYI} + trend_{FYI_m} \cdot C_{FYI}(+trend_{YI_m} \cdot C_{YI}))$$
With SIT the sea ice thickness, m the month you are correcting and C the partial concentration of each category. (Young ice trending is between parentheses as you not correct YI thickness) In other words do you make a weight average of the the SIT with the values of the trends and the partial concentration ?

L 182-184: I would also put this two lines within the conclusion maybe juste clarifying L 375 as you suggest that the proxy SIT need to be trended to take into account MYI thinning.

Figure 7 : I'm not sure that this figure is so convincing and enhance your dataset. The new series is consistent with CS2 which is very good. Altimetry products show big discrepancies with BGEP for the winters 2006/2007 and 2007/2008, so it doesn't surprise me that there are quite big differences for these years in this product too. Nevertheless it might be more meaningful to show a time series (still with CS-2), we would see that the seasonality is well represented too for other winter.

Section 4.3 : Maybe it would be even more readable at some point to put this section as a part of the discussion section as it both discusses the time series and the method and not only the method. But this is more a detail.

L 223: I may have missed this information, but I didn't understood how you estimated the 30-50 cm uncertainties of the product.

L 310-311: RSE characterized how the prediction fits the reference SIT (CS-2). Values are relative to CS-2 SIT not to the trend. Maybe I didn't get the point, why the variability between SIT proxy uncorrected and CS-2 SIT represent the variability to the trend, CS-2 SIT also get a variability to the trend isn't it ?

L 369 : « mean trend » ? This is still a bit confusing, I suppose it is the trend for the SIT for the whole studied area not the mean trend of each region.

L 375 : Which one ? Corrected or not, the not corrected will not provide consistent trends isn't it ?

Which kind of regression are you using to compute trends ?

---

## Author Comment (AC1)

**Response to referee comments – Marion Bocquet**

We thank the referee for their useful comments. We appreciate the time and effort dedicated to providing feedback on our manuscript and are grateful for the comments. We believe we have been able to address each of them.

**General comments:**

*1. The first one, relatively minor, concerning the representativity of the dataset, to what extent are the data used to train the NN over the period with CS-2 representative of the whole area you are trying to estimate, in other words, is there a bias in the sampling of the dataset that can lead to a bias in the predicted SIT. For instance, is the backscatter of CS-2 'area' is representative of CS-2+channels area ? A simple plot of the backscatter distribution in the region used for training and the global region would give a good idea, same for the parameters used from ice charts that allow to train the NN (floe width etc)*

Removing the CAA channels from the training dataset removes around 1/3 of the datapoints (depending on the month). Below is a plot (Fig 1) of the distribution of backscatter from C-band and Ku-band in November and April for the training region and the full region. As you can see the distribution is very similar.

[Figure]

Fig 1. Distribution of scatterometer backscatter

The same is true for the features used from the ice charts. Figure 2 shows an overview of some of them for April. The one feature where we would expect the difference between the full region and the training region to be strongest is for fast ice, as this is more common in the channels in the CAA, and less so in the open seas. However, even for this feature there is a full spread from 0 to 1 in the training region. The fact that the distribution of features in the training region is representative for the full region has been made clear with a few added sentences to the text.

[Figure]

Fig 2. Distribution in training region and full region of key features used from the ice charts in April.

2. The second one concerns the data « trending » that is made for MYI thinning correction. To what extent PIOMAS trend for MYI SIT is supposed to represent the real trend especially before 2010 as we usually see higher discrepancies while going back in time with PIOMAS ? This can maybe be discussed a bit more in the conclusion or in a discussion section ? Especially that, as explained by the authors, PIOMAS thick ice (MYI ?) is underestimated, linearly for the past 25 years ? How confident are the authors by « retrending » using PIOMAS ? I was also wondering it which extent does it make sens to trend the data and then to explain changes in the trend with the data as we could directly study PIOMAS dataset to get these trends ? I am a bit confused with the corrected and not corrected trends products. Why the authors detail trends and explain changes for the not corrected sea ice thickness product, as it is supposed not to reflect the trends ? Especially regarding the differences between not corrected and corrected trends for all regions and all months. Maybe in a way the study of the trends with the corrected dataset will be more relevant if it would have been validated before as the SIT changed, but I am not sure that is the purpose of this study. What bothers me is probably L 170, and

*that SIT are consistent but not to estimate trends… so the dataset is not so consistent as correcting the SIT by the trend will change the SIT values. Same for fig 11 and 12, the not corrected trend values are shown but they are supposed not to be so relevant. Could you make Fig 11/12 (maps) with the trends for corrected SIT ? Are the spatial patterns similar ? (Not necessarily in the manuscript for now.)*

We do agree with the reviewer that the way the two products were presented was confusing. We have followed the reviewers advise to do the validation on the corrected SIT proxy dataset, and we found that this gave better results than the validation on the not-corrected proxy SIT gave. This gives us more trust in our decision to apply this correction, despite the uncertainties that PIOMAS can bring. We have included a more thorough discussion of the uncertainties in PIOMAS, how the correction is exactly applied, and what the difference between the corrected and not-corrected product is and which one to use in which situation.

We have changed the maps of Figure 11 and 12 in the revised manuscript to show the SIT trends from the corrected SIT proxy. The spatial patterns are the same, the magnitude of the trend is stronger, and now negative almost everywhere.

**Specific comments**

*L2/3: You should be mentioned in the abstract that the estimation is based on Ice chart/scatterometers as it is one of the main elements of the thickness estimation.*

This has been added to a revised version of the manuscript.

*L6: 'mean trend' is a bit confusing. If I am not mistaking, this is not the mean trend this is the trend for the non corrected product for all the period and the whole area.*

There will be clarification added to the new manuscript that this mean represents the whole area.

*L128: why to chose the mean of the two product as you have seen that ku seems to be more more uncertain (higher RMSE)?*

Neither the Ku-band nor C-band scatterometer data are available over the whole time period. Because of this reason we cannot choose one of the two. Another possibility would have been to select C-band where available and fill in the gaps using Ku-band. However, as the two have different wavelengths, penetrate the snow and ice to a different depth and show different properties of the sea ice, we think this would have led to small inconsistencies. To remove some of this, we decided to take the mean where possible, and when only one of the two was available, select that one. Moreover, the different scatterometer sensor wavelengths have their own strengths and weaknesses, by combining both we get the full spread of their capabilities.

*L176: Just to be sure, are you trending the SIT as following :*

$$SIT_{corr_m} = SIT \cdot t \cdot \left( trend_{MYI_m} \cdot C_{MYI} + trend_{FYI_m} \cdot C_{FYI} \left( + trend_{YI_m} \cdot C_{YI} \right) \right)$$

*With SIT the sea ice thickness, m the month you are correcting and C the partial concentration of each category. (Young ice trending is between parentheses as you not correct YI thickness) In other*

*words do you make a weight average of the the SIT with the values of the trends and the partial concentration?*

> Yes, this is a much clearer way of stating this function, thank you. We will be using this phrasing of the correction function in the revised manuscript.

*L 182-184: I would also put this two lines within the conclusion maybe juste clarifying L 375 as you suggest that the proxy SIT need to be trended to take into account MYI thinning.*

> We have clarified this in the revised manuscript.

*Figure 7: I'm not sure that this figure is so convincing and enhance your dataset. The new series is consistent with CS2 which is very good. Altimetry products show big discrepancies with BGEP for the winters 2006/2007 and 2007/2008, so it doesn't surprise me that there are quite big differences for these years in this product too. Nevertheless it might be more meaningful to show a time series (still with CS-2), we would see that the seasonality is well represented too for other winter.*

> We have created a time series of the proxy SIT and the ULS moorings, included here:

[Figure]

> We do think this figure shows that the seasonal cycle is captured. However, we think this is already shown by figure 9 in the manuscript (fast ice thickness comparison). It is good to see that the seasonal cycle is captured by the proxy SIT product, however, we think it is also important to see if the interannual variability is captured by the proxy product. We aim to show this in Figure 7. We agree that there are years where the comparison with the ULS BGEP moorings shows large differences, but think these differences are explainable and that this figure is of added value to the manuscript. We therefore suggest to not change this figure in the main text. We will add the figure above to the Supplementary Materials.

*Section 4.3: Maybe it would be even more readable at some point to put this section as a part of the discussion section as it both discusses the time series and the method and not only the method. But this is more a detail.*

> We have chosen to not follow the results-discussion set up but instead have a chapter that presents and discusses the 'Model performance' and a chapter that presents and discusses the 'sea ice thickness proxy-product', so the outcome of the model. We do believe that section 4.3 (Limitations and potential), only discusses the limitations and potentials of the model (and not the proxy SIT record itself) and should thus belong in the chapter on model performance.

*L 223: I may have missed this information, but I didn't understood how you estimated the 30-50 cm uncertainties of the product.*

The 30-50 cm is the testing error of the model as presented in Figure 4. I do agree that this may not be exactly the same thing as the uncertainty of the product, as the testing error treats CryoSat-2 SIT as 'correct', and thus only shows the error in the model compared to CryoSat-2 SIT and not actual SIT. We have clarified in the revised manuscript that this is the testing uncertainty of the model.

*L 310-311: RSE characterized how the prediction fits the reference SIT (CS-2). Values are relative to CS-2 SIT not to the trend. Maybe I didn't get the point, why the variability between SIT proxy uncorrected and CS-2 SIT represent the variability to the trend, CS-2 SIT also get a variability to the trend isn't it?*

The residual standard error (RSE) does not look at how the predictions fit the CS-2 SIT. As stated in line 311, it characterises the standard deviation on the residuals of the regression model on the proxy SIT product. It quantifies how far the data points are scattered around the fitted regression line, and thus the variability from the trend. This is done according to:

$$RSE = \sqrt{\frac{\sum(y_i - \hat{y}_i)^2}{n - 2}}$$

Where $y_i$ is the observed value of mean SIT in the proxy SIT product for given year $i$, $\hat{y}_i$ is the expected value in the fitted linear regression model for the same year, and $n$ is the number of observations (years). This is different from the root-mean-square-error (RMSE) used to compare two datasets (like the proxy SIT product and the CS-2 observed SIT), used in other parts of the manuscript, which might have been the source of the reviewer's confusion. We have included this formula in the revised manuscript to clarify this.

*L 369: « mean trend »? This is still a bit confusing, I suppose it is the trend for the SIT for the whole studied area not the mean trend of each region.*

We have clarified that this is the mean trend over the full area in a revised version of the manuscript.

*L 375: Which one? Corrected or not, the not corrected will not provide consistent trends isn't it?*

The trends and variability in SIT can be studied by either of the products, the choice which one needs to be made on the basis of what someone wants to study. The not-corrected product can be used to study SIT trends caused by changes in ice type. However, this product does not include changes in SIT caused by the thinning of a given ice type, like MYI, which the corrected product does. We have clarified this in the revised manuscript.

*Which kind of regression are you using to compute trends?*

We are assuming linear trends and thus using a linear regression. This was clarified in a revised version of the manuscript.

---

## Author Comment (AC2)

**Response to referee comments – Robert Ricker**

We thank the referee for their useful comments. We appreciate the time and effort dedicated to providing feedback on our manuscript and are grateful for the comments. We believe we have been able to address each of them.

**Major comments:**

*1) I find there is a lack of information regarding the data and methods, especially scatterometer data, but also ice charts, partly "hidden" in the supplements. For example, I recommend being more specific on the used ASCAT data. Which data product have you used? The sigma-0 at 40 deg incidence? Moreover, information like the workflow diagram is key to understand the study and should be included in the paper. In general, I think information, crucial for the paper should be present in the actual paper, while Supplements only support the paper, e.g. additional plots that show something in more detail.*

> We agree with the reviewer that having this information in the Supplementary Materials is not as accessible as it should be. Therefore, we have added the workflow diagram into the main paper. However, we also think that having the full technical details on the scatterometer data is not critical to support the conclusion of the paper and it would disrupt the flow of description. Therefore, we think it's best to move the description of the scatterometer data and its consistency across missions to the appendices (which are printed in the same document as the manuscript).

*2) I wonder if the output can be improved by using different training data. The Beaufort Sea is known for being very challenging for satellite altimetry, because of the mixed ice types, low SIT correlation lengths, and sometimes high drift speeds. I suppose that this leads to some misfit between altimetry, scatterometry, and ice chart data. What about using only areas, where confidence in CS2 SIT is better?*

> The locations where training of the model is possible relies on the availability of the CIS ice charts, which are available in the Beaufort Sea, Baffin Bay, and the channels of the CAA. We believe the reliability of the CS2 dataset in the CAA is low, mainly owing to the lack of leads, and thus don't use this region in the training. If we were to remove the Beaufort Sea from the training dataset, we would only be left with Baffin Bay. This would give a rather small region to train on, with a mix of ice types lacking in diversity and therefore unrepresentative of the larger Canadian Arctic region we apply the model to. There is very little MYI ice in this region for example, so the training dataset would not have the same variability in the model features as the full region would have.
> We are aware of the fact that including the Beaufort Sea in the training dataset brings in uncertainties, but think that the addition of this region with a mix of ice types, sea ice thicknesses, and scatterometer results, will make the model better as it will learn more from the diverse inputs. We also think having the low spatial resolution we have (50 km), might decrease some of the uncertainty caused by high sea ice drift.
> We have added a discussion of these uncertainties in section 4.3.

*There are also CS2 SIT products that provide data in the Canadian Archipelago, for example the AWI product. It would be interesting to compare the proxy product with such estimates, too.*

> We have made a comparison to the AWI CS2 SIT product and included this in the revised supplementary materials and referred to it in the revised manuscript.

*3) I would expect a certain bias between scatterometry data sets from different sensors (but same frequency bands). For example, between ASCAT and ERS, because the sensors are different (as pointed out in the supplements). Can you rule out a bias that might affect derived trends in the proxy product? Or should it be at leased discussed in the limitations section?*

> We have looked at whether there was a bias when combining different scatterometers, including OSCAT-1 and OSCAT-2, OSCAT-1 and QuickScat, and ERS-2 and ASCAT. This was included in the Supplementary Materials but is now moved to Appendix A. For ERS-2 and ASCAT there is unfortunately no overlap in time, instead we looked at the PDFs of backscatter for both satellites for different ice ages. We see a similar backscatter in ERS-2 and ASCAT data for the same ice age and for this reason conclude that we did not need to apply a correction before merging the products in a C-band scatterometer product. We have now included a few more lines on this in the main text as well.

*4) A major concern is the correction to thinning of ice types (3.4). I understand the problem, but PIOMAS also comes with considerable uncertainties in the study regions. And it is a model, too. I wonder if it makes sense to correct the proxy product (based on a ML model) with trends from a sea ice model, while one could then also just take PIOMAS to look at trends. Figure 11 actually shows (except Baffin Bay may be) that the trend in the corrected product is basically coming from PIOMAS? I think it is feasible to compare with PIOMAS trends to discuss the problem. But I am not sure if this can be sold it as a separate product. I would argue that it is less confusing for potential users if there is just one product, where limitations and uncertainties are clear.*

> Following a comment from the other referee, we have now applied the validation to the independent SIT datasets on the corrected proxy SIT product as well (we did it on the non-corrected product before). We find that the correlation (for most of the locations and datasets) and anomaly correlation coefficient (for all datasets and locations) is higher after the ice type-thickness trend correction is applied. This makes us believe that despite the uncertainties present in the PIOMAS product, adding the correction is better than not adding the correction. We therefore choose to keep the ice type-sea ice thickness correction in the manuscript. We do agree that presenting the two different products was confusing, and therefore now focus on just the corrected version, as we've found this gave better results in the validation. We have included a more thorough discussion of the uncertainties of this correction in section 4.3.
>
> We don't think the adding of the ice type-SIT correction using PIOMAS results in the trends being so similar to PIOMAS that you could just take PIOMAS trends instead. The ice charts and scatterometer data used by the proxy SIT product contribute significantly to the trends and spatial variability. For example, in April over the full study area, PIOMAS gives an overall trend of -0.5 cm/yr for 1996-2020, whereas the corrected proxy product results in a trend of -1.5 cm/yr. Moreover, the spatial variability in this trend is large, with PIOMAS showing stronger trends than the proxy SIT in the Beaufort Sea and Parry Channel, and weaker trends compared to the proxy in Baffin Bay and the Arctic Ocean Periphery.

*5) The uncertainty estimate of 30-50 cm is based on the estimated model uncertainty, verified by the ULS comparison, if I understand correct. How do the OIB data compare to the proxy product in numbers? Section 4.2.2 is rather descriptive. It would be good if some numbers to verify the uncertainties can be presented here as well, like RMSD values etc.*

> We agree that this section is mostly based on descriptions and the visuals. We have now determined the correlation and RMSE between the OIB tracks and coinciding proxy SIT

values, and included these numbers in Figure 8. These numbers have also been added to section 4.2.2.

**Specific comments:**

*L81: I think the justification should be formulated here.*

We have included a few more lines on this justification in the main text.

*L122: Perhaps mention that scikit-learn is a python library.*

This will be added to the revised manuscript.

*L123: Any reasoning why 95?, and the value for the maximum depth? Is it an empirical choice, after trying different setups?*

These parameters were selected using the hyperparameter tuning function GridSearchCV in the python package scikit-learn. This function optimizes the parameters by using a cross-validated grid-search over a parameter grid.

*L136: May be consider introducing sub-sections for each validation data set.*

This was added to the revised manuscript.

*L147: Is it the same ice density as used for the CS2 SIT product? Why do you not distinguish between FYI and MYI density, see Alexandrov et al. (2010), Jutila et al. (2021)?*

Thank you for pointing this out. We have changed this to make the sea ice density different for FYI and MYI according to Alexandrov et al. (2010), in line with the CS2 SIT product.

*L149: I think it should be clarified that OIB does not measure SIT but uses the ATM laser and the snow radar to measure snow freeboard and snow depth and convert freeboard into thickness. This conversion goes along with several uncertainties as well. So, I suggest to rather write that OIB provides SIT "retrievals" or something similar.*

We have changed the wording to say SIT retrievals.

*L178: I suggest shortening the lower case terms here to improve readability of this formula. And what is the difference between the "i" and "icecategory"? Please clarify.*

We have changed the formulation of this formula to make it more intuitive.

*L191: May be shortly explain the "10-fold cross validation RMSE", what does it mean? And is the testing error directly related to the uncertainty given in the Abstract (30 to 50 cm)?*

An explanation of the cross validation was added to the revised manuscript. Yes, the testing error was directly related to the uncertainty in the abstract. We have now clarified that this is the testing error, not the full uncertainty.

*L199: How have you chosen the 20% CS2 data? Are they randomly picked points? Or did you cut out a certain area? This distinction can be important as the correlation between both chunks of CS2 data might be different. Please clarify in the text.*

Yes, these points were randomly selected. We will clarify this in the revised manuscript.

*L217: The usage of "proxy product" and "not corrected proxy product" is sometimes confusing. May be use a more consistent nomenclature, e.g. proxy_corr product and proxy_nocorr (or only proxy) product, throughout the paper.*

We have made this more consistent.

*Figure 2: I do not understand Fig. 2b) - what does "form of ice" mean here? And why does it have "km" as unit? Shall it relate to floe size? Or shall it show different forms of ice more in the sense of ice type, but then the colormap should be discrete?*

The form of ice comes from the ice chart's WMO egg code, and relates to floe size or kind of ice (e.g., iceberg, fast ice). The forms of ice are defined using a minimum width, described in Table 'Coding for Forms of Ice' on https://www.canada.ca/en/environment-climate-change/services/ice-forecasts-observations/publications/interpreting-charts/chapter-1.html. What we did for this figure was to take these widths for each of the present forms of ice in a polygon, and weight them according to their relative partial concentrations, into a final number that described the weighted mean for present form of ice. This number is a bit subjective and not used anywhere else, only to depict the spatial variability in forms of ice. We do agree that this is confusing, and have changed Figure 2b to state forms of ice – small to large, so it is more consistent with Figure 2a.

*Figure 8: The histograms are very difficult to separate. Maybe you can have the columns next to each other (like in the supplements) and/or use colors that are easier to separate.*

This was changed in the revised manuscript to having the columns next to each other.

*Figure 9: It is very difficult to compare the in-situ values, as the time axis is very coarse, while the interannual gradient is quite strong.*

We have added a subfigure for each location showing the mean seasonal cycle; the subfigure is wide enough to compare the two.

*Figure 11/12: For the line plots, it would help to have legends. Why are you using colors for the different regions if you present them in different boxes? I am also slightly confused with the "solid and weaker circles". "Colours show the trend for the not corrected version" -> But they are all colored? This figure not so easy to read.*

We have removed the different colours and made all the line plots black. We have also added a legend to the first plot and updated the caption. We hope this makes the figure easier to read.

*Figure S1: I suggest including this figure in the paper, as it shows the workflow to derive the proxy product - too crucial for the Supplements from my point of view.*

This figure is now included in a revised version of the manuscript.

---

## Author Response (AR1)

Dear Yevgeny Aksenov,

Thank you for the consideration and your time as editor. We have revised the manuscript according to the comments by the referees and our responses to these comments. The main changes regard the shown results now being consistently the same corrected proxy SIT product, an extended discussion on the ice type-sea ice thickness correction using PIOMAS, and the move of the scatterometer data section to the Appendix.

An additional change to the manuscript is that we decided to remove the CAA regions from the PIOMAS ice type-thickness correction, which results in slightly different results in the proxy SIT product and trends.

We hope these revisions are satisfactory.

Kind regards,
Isolde Glissenaar

---

## Author Response (AR2)

Dear Yevgeny Aksenov,

Thank you for accepting the manuscript and for your work as editor.

Kind regards,

Isolde Glissenaar

Reply to specific comments

L 143 : Linear trend, thanks for adding this precision. I will reformulate my question as I think it was not clear. I was wondering which estimation method was used. Indeed, there is a lot of variability in these regions and usual linear regression made with python/scipy used least squared method so trends can biased a lot due to outliers. I would not suggest to change the trends computation but to precise this choice because climate trends are usually computed with estimators that gives less weight to outliers (e.g. Theil-Sen algorithm), same for the test which is used to estimate the significance of the trend.
*We have clarified that it is indeed the least-squares method that was used.*

Figure 4: If ever it's possible to enlarge the font or the figure a little to make it easier to read, that would be great.
*It is not really possible to enlarge the font size and keep the figure in the same dimensions. We hope it is possible to have this figure larger than the 12cm in the TC figure guidelines, or to have the figure in Landscape in the final document.*

Abstract: May be the use of PIOMAS should be mentioned in the Abstract, since it plays a significant role in the paper?
*We have added this to the abstract.*

L143: "we retrieved the linear trend in PIOMAS mean thickness": The PIOMAS mean thickness as provided by PSC represents effective ice thickness (Volume per unit Area). Is this consistent with the proxy-product (and eventually the CryoSat-2 SIT)? Please clarify if needed.
*We have corrected the PIOMAS effective thickness to real thickness using the PIOMAS sea ice concentration.*